# Model-based Safe Deep Reinforcement Learning via a Constrained Proximal Policy Optimization Algorithm

**Ashish Kumar Jayant**
Department of Computer Science Automation
Indian Institute of Science
Bangalore
ashishjayant@iisc.ac.in

**Shalabh Bhatnagar**
Department of Computer Science Automation
Indian Institute of Science
Bangalore
shalabh@iisc.ac.in

## Abstract

During initial iterations of training in most Reinforcement Learning (RL) algorithms, agents perform a significant number of random exploratory steps. In the real world, this can limit the practicality of these algorithms as it can lead to potentially dangerous behavior. Hence safe exploration is a critical issue in applying RL algorithms in the real world. This problem has been recently well studied under the Constrained Markov Decision Process (CMDP) Framework, where in addition to single-stage rewards, an agent receives single-stage costs or penalties as well depending on the state transitions. The prescribed cost functions are responsible for mapping undesirable behavior at any given time-step to a scalar value. The goal then is to find a feasible policy that maximizes reward returns while constraining the cost returns to be below a prescribed threshold during training as well as deployment.

We propose an On-policy Model-based Safe Deep RL algorithm in which we learn the transition dynamics of the environment in an online manner as well as find a feasible optimal policy using the Lagrangian Relaxation-based Proximal Policy Optimization. We use an ensemble of neural networks with different initializations to tackle epistemic and aleatoric uncertainty issues faced during environment model learning. We compare our approach with relevant model-free and model-based approaches in Constrained RL using the challenging Safe Reinforcement Learning benchmark - the Open AI Safety Gym. We demonstrate that our algorithm is more sample efficient and results in lower cumulative hazard violations as compared to constrained model-free approaches. Further, our approach shows better reward performance than other constrained model-based approaches in the literature.

## 1 Introduction

Deep Reinforcement Learning has provided exceptional results both in the case of discrete action settings [Mnih et al., 2013] as well as continuous action domains such as locomotion tasks [Haarnoja et al., 2018], [Schulman et al., 2017], [Schulman et al., 2016]. However, most of the RL algorithms perform significant number of random exploratory steps during learning as well as deployment which can lead to agents performing undesirable and hazardous behaviour. This limits the application of RL algorithms in the real world. In scenarios like robot navigation [Han et al., 2018], autonomous driving [Kendall et al., 2018], healthcare [Yu et al., 2020], etc., where RL has potential applications, unsafe behavior can have hazardous consequences even on human life and property.

In García et al. [2015], the authors have provided a comprehensive survey of several notions of safety and the associated problem formulations. In our work, we focus on a constraint-based notion of safety. In Constraint-based RL, the goal is to maximize long-term expected reward returns and keep the expected cost-returns below a prescribed threshold. This problem is known as the Safe

36th Conference on Neural Information Processing Systems (NeurIPS 2022).

Exploration problem. In [Ray et al., 2019], the authors advocate that safety specifications should be separate from task performance specifications. It also helps in formulating Safe Exploration as a Constrained Optimization Problem and methods used in the optimization literature [Bertsekas, 1996] can be used to solve this problem. Constrained Markov Decision Process (CMDP) [Altman, 1998] provides a framework to keep task performance specifications and safety specifications separate from one another. Here in addition to single-stage rewards, state transitions receive single-stage costs as well. The prescribed cost functions are responsible for mapping undesirable behavior at any given time-step to a non-negative scalar value.

The existing model-free algorithms used in the Constrained-RL setting suffer from low sample efficiency in terms of environment interactions, i.e., these algorithms require large number of environment interactions (to converge) that in turn would lead to a large number of hazardous actions due to unsafe exploration. This serves as the motivation for us to use a Model-based approach for Constrained-RL.

**Our Contributions:** We propose a simple and sample efficient model-based approach for Safe Reinforcement Learning which uses Lagrangian relaxation to solve the constrained RL problem. We highlight the issue that arises due to the use of truncated horizon in Constrained RL and suggest a way to incorporate that in our setting. We demonstrate that our approach is $\sim 3 - 4$ times more sample efficient than its analogue Model-free Lagrangian relaxation approach and reduces its cumulative hazard violations by $\sim 60\%$. Our approach also outperforms model-free Constrained Policy Optimization (CPO) [Achiam et al., 2017] in terms of constraint satisfaction. We also compare our approach with model-based approach [Sikchi et al., 2021a] and we observe that our approach does better in terms of reward performance and has competitive cost performance as well.

## 2 Related Work

Different notions of safety and their mathematical formulations are provided in García et al. [2015]. There are several works on the lines of formulating the Safe RL problem in the setting of CMDP. In some of the early works, an actor-critic algorithm for CMDP under the long-run average cost criterion is proposed in Borkar [2005] that is however for the case of full state representations. Actor-critic algorithms with linear function approximation have been proposed in Bhatnagar and Lakshmanan [2012], Bhatnagar et al. [2013] for the long-run average cost setting and in Bhatnagar [2010] for the infinite horizon discounted cost scenario. The procedure in the aforementioned references involved forming a Lagrangian by relaxing the constraints. The algorithms in these papers are based on multi-timescale stochastic approximation with updates of the Lagrange parameter performed on the slow timescale, the policy updates on the medium timescale and the updates of the value function (for a given policy) performed on the fast timescale.

In more recent work, in Achiam et al. [2017], a trust region based constrained policy optimization (CPO) framework is proposed, which involved approximation of the problem using surrogate functions for both the objective and the constraints and included a projection step on policy parameters that needed backtracking line search, making it complicated and time-consuming. CPO showed near constraint satisfaction in every iteration of the policy updates in standard Mujoco environments modified for safe exploration [Achiam et al., 2017] but didn't yield a constraint satisfying policy in challenging Safe RL benchmark Safety Gym [Ray et al., 2019]. Another work [Yu et al., 2019] involved using surrogate functions for approximation of both objective and constraint functions. Their procedure involved constructing a sequence of convex optimization problems for which they showed that the sequence of stationary points converges to the stationary point of the original non-convex problem.

In Ray et al. [2019], Lagrangian relaxation of the Constrained RL problem is used and combined with PPO [Schulman et al., 2017] to give a PPO-Lagrangian algorithm and with TRPO [Schulman et al., 2015] to give a TRPO-Lagrangian algorithm. These algorithms were seen to outperform CPO [Achiam et al., 2017] in terms of constraint satisfaction on several environments in Safety Gym. Also, these algorithms are simpler to implement and tune. Another Lagrangian-based method, see Tessler et al. [2018], used a penalized reward function for optimizing their agent and showed convergence to optimal feasible policies using a two-timescale stochastic approximation scheme where the Lagrange multiplier is updated on a slower timescale as compared to the policy parameters as was the case with Borkar [2005], Bhatnagar [2010], Bhatnagar and Lakshmanan [2012]. In Zhang et al. [2020], the authors proposed a first order constrained policy optimization (FOCOPS) method that involved

solving the optimization problem in a non-parametric space and then projecting it back into the parametric space. Approaches in Tessler et al. [2018] and Zhang et al. [2020] performed poorly on Safety Gym. In Stooke et al. [2020], a PID-based approach to damp oscillations in Lagrangian methods is proposed, which is seen to minimize constraint violations. In Wen and Topcu [2021], the Cross Entropy method [de Boer et al., 2005] is used for finding a safe policy and convergence using the ODE method is shown, however, empirical results are presented only on a primitive and less challenging environment. In Dalal et al. [2018], the authors formulated a state-wise constrained policy optimization problem where at each transition a constraint needs to be satisfied and an analytical method for correcting the unsafe action using a safety layer trained using random exploration was proposed. In Chow et al. [2018], the authors proposed constructing Lyapunov functions to guarantee safety of the behaviour policy under a CMDP framework. In Chow et al. [2019], the above Lyapunov based method was extended to continuous control but it's performance in Safety Gym environment [Ray et al., 2019] was not good in terms of rewards obtained [Sikchi et al., 2021b]. In Liu et al. [2022], the method involved uses the tangent space of the constraint manifold to learn a safe policy but the approach is seen to be highly specific to the environment used.

For unconstrained state-of-the-art Model-based RL algorithms in Deisenroth and Rasmussen [2011], Luo et al. [2018], Chua et al. [2018], Kurutach et al. [2018], Heess et al. [2015], a comparison of the empirical performance on Open AI Gym [Brockman et al., 2016] is shown in Wang et al. [2019]. Also, Sikchi et al. [2021a] propose augmenting the planning trajectory with terminal value function to incorporate long-horizon reasoning in model-based methods, since the approaches in Deisenroth and Rasmussen [2011], Luo et al. [2018], Chua et al. [2018], Kurutach et al. [2018], Heess et al. [2015] plan over a fixed and short horizon to avoid aggregation of error. This requires good approximation of both model as well as value function.

There are also several works that use model-based RL to tackle the problem of safety. In Liu et al. [2020], a model based approach is proposed to learn the system dynamics and cost model. Then roll-outs from the learned model are used to optimize the policy using a modified cross-entropy based method which involves sampling from a distribution of policies, sorting sample policies based on constraint functions and using them to update the policy distribution. However, their implementation involves a data collection step using random policy for large number of episodes which itself is risky in real-world scenarios. In Cowen-Rivers et al. [2020], model dynamics is learned using PILCO [Deisenroth and Rasmussen, 2011] and instead of the discounted cost constraint function, conditional value at risk (CVaR) based constraint function is used [Uryasev and Rockafellar, 2001, Chow and Ghavamzadeh, 2014]. In Thomas et al. [2022], penalized reward functions are used instead of a separate cost function, then model of the environment is learned and the soft-actor critic algorithm [Haarnoja et al., 2018] is used to optimize the policy. In this approach safe and unsafe states are also needed to be specified upfront.

## 3 Background

### 3.1 Constrained Markov Decision Process (CMDP)

A CMDP is denoted by the tuple $(S, A, R, C, \gamma, \mu)$ where $S$ denotes the state space, $A$ is the action space, $R : S \times A \times S \to \mathbb{R}$ is the single-stage reward function, $C : S \times A \times S \to \mathbb{R}$ denotes the associated single-stage cost function (we assume a single constraint function for simplicity here), $\gamma$ is the discount factor and $\mu$ signifies the initial state distribution. We assume that both $R$ and $C$ are non-negative functions.

By a policy $\pi = \{\pi_0, \pi_1, \ldots\}$, we mean a decision rule for selecting actions. It is specified as follows:

For any $k \geq 0$ and $s \in S$, $\pi_k(s) \in \mathbb{P}(s)$ is the probability distribution $\pi_k(s) \stackrel{\triangle}{=} (\pi_k(s, a), a \in A(s))$ where $\pi_k(s, a)$ is the probability of picking action $a$ in state $s$ at instant $k$ under policy $\pi$. In the above, $A(s)$ is the set of feasible actions in state $s$ and so $A = \cup_{s \in S} A(s)$. Such a policy is also often referred to as a randomized policy. A stationary policy is a randomized policy as above except with $\pi_k = \pi_l, \forall k \neq l$. Thus, a stationary policy selects actions according to a given distribution regardless of the instant when an action is chosen according to the given policy. By an abuse of notation, we denote a stationary policy as $\pi$ itself.

We shall consider here a class of stationary policies $\pi_\theta$ parameterized by a parameter $\theta$. Our objective function is defined via the infinite horizon discounted reward criterion where for a given $\pi_\theta$ we have

$$J^R(\pi_\theta) = \mathbb{E}\left[\sum_{t=0}^{\infty} \gamma^t R(s_t, a_t, s_{t+1}) \mid s_0 \sim \mu, a_t \sim \pi_\theta, \forall t\right]. \tag{1}$$

The (cost) constraint function is similarly specified via the following infinite horizon discounted cost:

$$J^C(\pi_\theta) = \mathbb{E}\left[\sum_{t=0}^{\infty} \gamma^t C(s_t, a_t, s_{t+1}) \mid s_0 \sim \mu, a_t \sim \pi_\theta, \forall t\right]. \tag{2}$$

Then $J^R(\pi_\theta), J^C(\pi_\theta) \in \mathbb{R}$. Let $d > 0$ denote a prescribed threshold below which we want $J^C(\pi_\theta)$ to lie. The constrained optimization problem then is the following:

$$\max_\theta J^R(\pi_\theta) \text{ s.t. } J^C(\pi_\theta) \leq d. \tag{3}$$

A parameter $\theta$ will be called a feasible point if the cost constraint is satisfied for $\theta$, i.e., $J^C(\pi_\theta) \leq d$.

### 3.2 Lagrangian Relaxation based Proximal Policy Optimization

The Lagrangian of the constrained optimization problem (3) can be written as follows:

$$L(\theta, \lambda) = J^R(\pi_\theta) - \lambda(J^C(\pi_\theta) - d), \tag{4}$$

where $\lambda \in \mathbb{R}^+$ is the Lagrange multiplier and is a positive real number. In terms of the Lagrangian, the goal is to find a tuple $(\theta^*, \lambda^*)$ of the policy and Lagrange parameter such that

$$L(\theta^*, \lambda^*) = \max_\theta \min_\lambda L(\theta, \lambda). \tag{5}$$

Solving the max-min problem as above is equivalent to finding a global optimal saddle point $(\theta^*, \lambda^*)$ such that $\forall(\theta, \lambda)$, the following holds:

$$L(\theta^*, \lambda) \geq L(\theta^*, \lambda^*) \geq L(\theta, \lambda^*). \tag{6}$$

We assume that $\theta$ refers to the parameter of a Deep Neural Network, hence finding such a globally optimal saddle point is computationally hard. So our aim is to find a locally optimal saddle point which satisfies (6) in a local neighbourhood $H_{\epsilon_1, \epsilon_2}$ which is defined as follows:

$$H_{\epsilon_1, \epsilon_2} \triangleq \{(\theta, \lambda) \mid \|\theta - \theta^*\| \leq \epsilon_1, \|\lambda - \lambda^*\| \leq \epsilon_2\}, \tag{7}$$

for some $\epsilon_1, \epsilon_2 > 0$. Assuming that $L(\theta, \lambda)$ is known for every $(\theta, \lambda)$ tuple, a gradient search procedure for finding a local $(\theta^*, \lambda^*)$ tuple would be the following:

$$\begin{align}
\theta_{n+1} &= \theta_n - \eta_1(n)\nabla_{\theta_n}(-L(\theta_n, \lambda_n)), \tag{8} \\
&= \theta_n + \eta_1(n)[\nabla_{\theta_n}J^R(\pi_\theta) - \lambda_n\nabla_{\theta_n}J^C(\pi_\theta)], \tag{9} \\
\lambda_{n+1} &= [\lambda_n + \eta_2(n)\nabla_{\lambda_n}(-L(\theta_n, \lambda_n))]_+, \tag{10} \\
&= [\lambda_n - \eta_2(n)(J^C(\pi_\theta) - d)]_+. \tag{11}
\end{align}$$

Here $[x]_+$ denotes $\max(0, x)$. This operator ensures that the Lagrange multiplier remains non-negative after each update. In (9)-(11), $\eta_1(n), \eta_2(n) > 0 \, \forall n$ are certain prescribed step-size schedules. We assume that the step-sizes $\eta_1(n), \eta_2(n), n \geq 0$ satisfy the regular step-size conditions. Thus, for $i = 1, 2, \sum_k \eta_i(n) = \infty, \sum_k \eta_i^2(n) < \infty$. Note however that $J^R(\pi_\theta)$ and $J^C(\pi_\theta)$ as specified in (1)-(2) are not a priori known quantities and need to be estimated. We discuss this in detail below.

### 3.2.1 Estimation

We run each episode for $T$ time steps in our experiments. Let $r_{t+1} \equiv R(s_t, a_t, s_{t+1})$ and $c_{t+1} \equiv C(s_t, a_t, s_{t+1})$, respectively, for simplicity. For each sample path we would have both a reward return as well as a cost return. Let $\hat{R}_t$ (resp. $\hat{C}_t$) be the reward-to-go (resp. cost-to-go) estimate. We compute $\hat{R}_t$ and $\hat{C}_t$ according to: $\hat{R}_t = \sum_{k=0}^{T-t-1} \gamma^k r_{t+k+1}, \hat{C}_t = \sum_{k=0}^{T-t-1} \gamma^k c_{t+k+1}$, respectively.

We use a neural network parameterized by $\psi_r$ to estimate reward signal based value function $V_{\psi_r}^R$ and a neural network parameterized by $\psi_c$ to estimate a cost signal based value function $V_{\psi_c}^C$. We run our simulations on $N$ parallel workers and then sample a mini-batch $\mathcal{M}$ of size $M \leq NT$ [Schulman et al., 2017]. We use mean-squared loss to estimate value functions on sampled mini-batches as follows:

$$Loss(\psi_r) = \frac{1}{MT} \sum_{\tau \in \mathcal{M}} \sum_{t=0}^{T} (V_{\psi_r}^R(s_t) - \hat{R}_t)^2, \tag{12}$$

$$Loss(\psi_c) = \frac{1}{MT} \sum_{\tau \in \mathcal{M}} \sum_{t=0}^{T} (V_{\psi_c}^C(s_t) - \hat{C}_t)^2. \tag{13}$$

Let $A_t^R$, $t \geq 0$, and $A_t^C$, $t \geq 0$, respectively, denote the advantage estimates w.r.t reward and cost value functions on the sample path. We compute them using Generalized Advantage Estimation [Schulman et al., 2016] to balance the bias and variance tradeoff of advantage estimates. We can have a range of advantage estimates as under.

$$A_t^1 = r_{t+1} + \gamma V(s_{t+1}) - V(s_t), \tag{14}$$
$$A_t^2 = r_{t+1} + \gamma r_{t+2} + \gamma^2 V(s_{t+2}) - V(s_t), \tag{15}$$
$$A_t^k = r_{t+1} + \cdots + \gamma^{k-1} r_{t+k} + \gamma^k V(s_{t+k}) - V(s_t), \tag{16}$$

for $k > 2$. The advantage estimate in (14) will have high bias but low variance while estimates in (16) with a higher value of $k$ generally have high variance but low bias. Let $\delta_t^R = r_{t+1} + V_{\psi_r}^R(s_{t+1}) - V_{\psi_r}^R(s_t)$ and $\delta_t^C = c_{t+1} + V_{\psi_c}^C(s_{t+1}) - V_{\psi_c}^C(s_t)$, respectively, denote the reward and cost temporal differences. Let $\bar{\lambda}$ be a parameter which adjusts the bias-variance tradeoff. Generalized Advantage Estimates [Schulman et al., 2016] for $A_t^R$, $A_t^C$ are then given by

$$A_t^R = \sum_{l=0}^{k} (\gamma \bar{\lambda})^l \delta_{t+l}^R, \tag{17}$$

$$A_t^C = \sum_{l=0}^{k} (\gamma \bar{\lambda})^l \delta_{t+l}^C, \tag{18}$$

respectively. Now we use PPO clipped objectives [Schulman et al., 2017] for estimation of $J^R(\pi_\theta)$, $J^C(\pi_\theta)$ as follows:

$$J^R(\pi_\theta) = \mathbb{E}_t[\min(r_t(\theta) A_t^R, \text{clip}(r_t(\theta), 1-\epsilon, 1+\epsilon) A_t^R)], \tag{19}$$

$$J^C(\pi_\theta) = \mathbb{E}_t[\min(r_t(\theta) A_t^C, \text{clip}(r_t(\theta), 1-\epsilon, 1+\epsilon) A_t^C)], \tag{20}$$

where $r_t(\theta) = \frac{\pi_\theta(a_t|s_t)}{\pi_{\theta_{old}}(a_t|s_t)}$ is the ratio of the probability of selecting action $a_t$ in state $s_t$ under parameter $\theta$ as opposed to $\theta_{old}$. Further, $A_t^R$ and $A_t^C$ are the estimated advantages based on the reward and cost returns, respectively, by time $t$ (see above) and $\epsilon$ is the clip-ratio which clips $r_t(\theta)$ to $(1-\epsilon)$ if it is less than $(1-\epsilon)$ and clips to $(1+\epsilon)$ if it is greater than $(1+\epsilon)$. This algorithm restricts the policy parameters to not change significantly between two iterations which helps in avoiding divergence. This approach is referred to as PPO-Lagrangian [Ray et al., 2019].

### 3.3 Model-based Constrained RL

We formulate a Constrained RL problem (21) using a model-based framework as follows:

$$\max_{\pi_\theta \in \Pi_\theta} J_m^R(\pi_\theta) \text{ s.t. } J_m^C(\pi_\theta) \leq d, \text{ where} \tag{21}$$

$$J_m^R(\pi_\theta) = \mathbb{E}\left[\sum_{t=0}^{\infty} \gamma^t R(s_t, a_t, s_{t+1}) \mid s_0 \sim \mu, \ s_{t+1} \sim P_\alpha(.|s_t, a_t), \ a_t \sim \pi_\theta, \forall t\right], \tag{22}$$

$$J_m^C(\pi_\theta) = \mathbb{E}\left[\sum_{t=0}^{\infty} \gamma^t C(s_t, a_t, s_{t+1}) \mid s_0 \sim \mu, \ s_{t+1} \sim P_\alpha(.|s_t, a_t), a_t \sim \pi_\theta, \forall t\right]. \tag{23}$$

In the above, $P_\alpha(.|s_t, a_t)$ is an $\alpha$-parameterized environment model, $d_i$ is a human prescribed safety threshold for the $i$th constraint and $\Pi_\theta$ is the set of all $\theta-$parameterized stationary polices. Note that we assume the initial state $s_0$ is sampled from the true initial state distribution $\mu$ and then $s_{t+1} \sim P_\alpha(.|s_t, a_t), \forall t > 0$. We would use approximation of environment $P_\alpha$ to create 'imaginary' roll-outs to estimate the reward and cost returns required for policy optimization algorithms.

## 4 Challenges in Environment Model Learning

In this section we discuss the challenges that commonly arise due to model learning in RL. We further highlight the challenge that arises from using environment model approximation in Safe RL settings.

1. **Handling aleatoric and epistemic uncertainties** : *Aleatoric Uncertainty* refers to the notion of natural randomness in the system which leads to variability in outcomes of an experiment. This uncertainty is irreducible because it is a natural property of the system. Hence in such cases, giving measure of uncertainty in model's prediction is a good practice. In Lakshminarayanan et al. [2017], Chua et al. [2018] uncertainty-aware neural networks are used which give an idea about uncertainty in prediction as well. They learn Gaussian distribution parameterized by neural networks. *Epistemic Uncertainty* refers to the notion of lack of sufficient knowledge in the model as a result of which the model does not generalize. In Lakshminarayanan et al. [2017], an ensemble of uncertainty aware neural networks with different initializations is proposed to reduce epistemic uncertainty for fixed data.

   For the learning environment model, we also use an ensemble of $n$ neural networks with random initialization. Each neural network's output parameterizes a multivariate normal distribution with diagonal covariance matrix. Suppose the $i$th neural network in the ensemble is parameterized by $\alpha_i$ and the mean and standard deviation outputs are $\mu_{\alpha_i}$ and $\sigma_{\alpha_i}$ respectively. Recall now that if a random vector $X \in \mathbb{R}^d$ is distributed according to the multivariate normal distribution parameterized by $(\mu, \Sigma_{d \times d})$ where $\mu \in \mathbb{R}^d$ is the mean vector and $\Sigma_{d \times d}$ is a $d \times d$ covariance matrix, then the probability density of $X$ is defined as, $P(x) = \frac{1}{(2\pi)^{d/2}} |\Sigma|^{-1/2} exp(-\frac{1}{2}(x-\mu)'\Sigma^{-1}(x-\mu)), x \in \mathbb{R}^d$.

   As a choice of loss function we use the negative log-likelihood loss for minimization (i.e., minimizing negative log of $P(X)$). For the $i$th neural network parameterized by $\alpha_i$, the loss function $L(\alpha_i)$ is given as follows:

$$L(\alpha_i) = \sum_{t=1}^{M} [\mu_{\alpha_i}(s_t, a_t) - s_{t+1}]^T \Sigma_{\alpha_i}^{-1}(s_t, a_t)[\mu_{\alpha_i}(s_t, a_t) - s_{t+1}] + \log |\Sigma_{\alpha_i}(s_t, a_t)|, \quad (24)$$

   where $\mu_{\alpha_i}(s_t, a_t)$ is the mean vector output of the $i$th neural network and $\Sigma_{\alpha_i}(s_t, a_t)$ is the covariance matrix which is assumed to be a diagonal matrix. Note that, using $n$ neural networks with random initialization tends to have a regularization effect. Intuitively, it introduces diversity in learned models to deal with more possible trajectories. Also, it has been shown empirically that with an increase in the number of models, the performance tends to improve (See section 6.3 and Figure 4 of Kurutach et al. [2018]), but increase in number of models also leads to increase in space complexity.

2. **Aggregation of Error** : In model-based RL, as we move forward along the horizon, the error due to approximation starts aggregating and predictions from the approximated model tend to diverge significantly from the true model. In order to tackle this problem, most of the model-based RL approaches [Deisenroth and Rasmussen, 2011, Kurutach et al., 2018, Janner et al., 2019] use shorter (or truncated) horizon during the policy optimization phase and achieve similar performance as Model-Free RL approaches. We use truncated horizon in our approach.

3. **Implication of using truncated horizon in Constrained RL** : When we use truncated horizon in Constrained RL, it leads to underestimation of cost returns (23) under the current policy and we use the prescribed constraints to threshold the returns obtained. This can lead to constraint violations in the real-environment where the cost objective is based on the infinite horizon cost return. We propose a hyperparameter-based approach to deal with this problem in the next section.

## 5 Model-based PPO Lagrangian

We propose a model-based algorithm obtained from relaxing the Lagrangian (see Algorithm 1) which alleviates the problem with obtaining a large number of samples in model-free Lagrangian based approaches and as a consequence also decreases the cumulative hazard violations. It is difficult to evaluate the policy without interacting with the real environment accurately.

---

**Algorithm 1** Our Approach: Model-based PPO-Lagrangian

---

1: **Input:** Initialize actor neural net parameter $\theta_0$, critic parameters $\psi_{r0}, \psi_{c0}$, ensemble models $[P_{\alpha_i}]_{i=1}^n$, Lagrange parameter $\lambda_0 \geq 0$, cost threshold = $d$, Environment Horizon = $T$, Model Horizon = $H$
2: **for** $i = 1, \ldots, N$ training epochs **do**
3:     Collect data tuples $(s_t, a_t, s_{t+1})$, for the $i$th ensemble using policy $\pi_{\theta_i}$, $i = 1, \ldots, n$, in the environment for $T$ time steps over multiple ($|E|$) episodes
4:     Train $[P_{\alpha_i}]_{i=1}^n$ by minimizing (24) w.r.t $\alpha_i$, $\forall i = 1$ $to$ $n$
5:     **while** Performance ratio $> PR_{threshold}$ **do**
6:         $s_0 \sim \mu$
        *{Note: For first pass we use a mix of real data and imaginary roll-out data. (See Appendix A)}*
7:         Collect data roll-outs as $a_t \sim \pi_{\theta_i}(\cdot|s_t)$, $s_t \sim P_{\alpha_q}(\cdot|s_t, a_t)$ (At each time step 'q' is randomly selected from $1, \ldots, n$) for $H$ time steps ($H < T$)
8:         Compute $J_{sample}^C(\pi_{\theta_t}) = \frac{1}{|E|} \sum_{p=1}^H \gamma^p C(s_t, a_t)$ where $|E|$ is the number of episodes
9:         Compute advantage, cost-advantage using (17) and (18) respectively
10:        Update $\lambda$ by replacing $J^C(\pi_\theta)$ with $J_{sample}^C(\pi_{\theta_i})$ and using an appropriate value of $\beta$ in (26).
        *{Multiple gradient updates for actor and critic}*
11:        **for** $k = 1, \ldots, K$ **do**
12:           Compute $J^R(\pi_{\theta_k}), J^C(\pi_{\theta_k})$ as in (19) and (20) respectively.
13:           Update parameters $\theta_k$ using (9)
14:           Update critic parameters $\psi_r$ and $\psi_c$ minimizing (12) and (13) respectively.
15:        **end for**
16:        Compute Performance Ratio ($PR$) using (25)
17:     **end while**
18: **end for**

---

For this we compute the Performance Ratio ($PR$) metric using ensemble models (see Kurutach et al. [2018]) that is defined as follows:

$$PR = \frac{1}{n} \sum_{i=1}^n \mathbb{1}(\zeta^R(\alpha_i, \theta_t) > \zeta^R(\alpha_i, \theta_{t-1})), \tag{25}$$

where $\zeta^R(\alpha_i, \theta_t) = \sum_{t=0}^T \gamma^t R(s_t, a_t, s_{t+1})$, $s_0 \sim \mu$ and $\forall t \geq 0 : s_{t+1} \sim P_{\alpha_i}(\cdot|s_t, a_t), a_t \sim \pi_{\theta_t}(\cdot|s_t)$. This measures the ratio of the number of models in which policy is improved to the total number of models in ensemble ($n$). If $PR > PR_{threshold}$, we continue training using the same model, if not then we break and re-train our environment model on data collected from the new update policy.

Another challenge that we encounter is the underestimation of $J^C(\pi_\theta)$ resulting from using a truncated horizon (of length $H$ in step 7 of Algorithm 1) to reduce the aggregation of error.

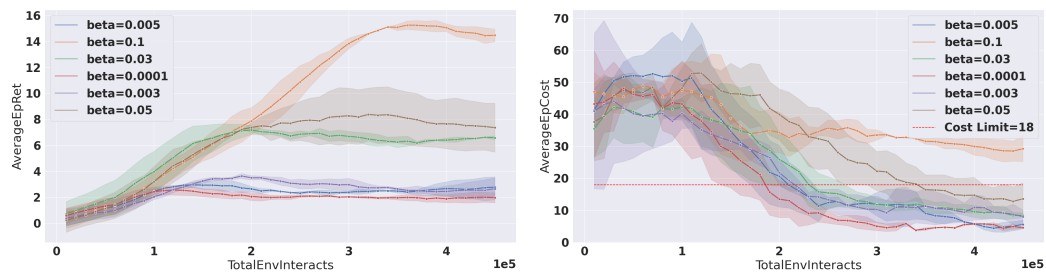

Figure 1: Effect of beta parameter $(\beta)$ on expected cost returns (left) and expected reward returns (right) in PointGoal environment. (Here $\beta = 0.1$ corrresponds to $\frac{H}{T}$)

.

This can potentially lead to constraint violating policies. So we need to make the safety threshold $(d)$ stricter. It might seem that changing this threshold $(d)$ proportional to the truncated horizon, i.e., setting $d := d * \frac{H}{T}$ might work better where $H$ is the truncated horizon and $T$ is the original environment horizon, but we found that this leads to constraint violations as well because we are learning policy using data from the approximated environment model that would bring in errors. Hence the cost estimate of the policy (Step 8 in Algorithm 1) is error-prone as well. To tackle this issue, we make safety threshold stricter using a hyperparameter $0 \leq \beta < 1$ by modifying the Lagrange multiplier update as follows:

$$\lambda_n = [\lambda_n - \eta_2(n)(J^C(\pi_\theta) - d * \beta)]_+. \tag{26}$$

The variation of expected cost returns and reward returns with respect to $\beta$ is shown in Figure 1 on Safety gym *PointGoal* [Ray et al., 2019] environment. We can observe that as we reduce $\beta$, the expected cost returns reduce because cost limit becomes stricter but choosing too small a $\beta$ leads to low reward returns as well.

## 6 Experimental Details and Results

We test our approach on Safety Gym environments [1] - *PointGoal* and *CarGoal* with modified state representations as used in other model-based Safe RL approaches [Liu et al., 2020, Sikchi et al., 2021a] that are more helpful in model-learning. We increase the difficulty of *PointGoal* and *CarGoal* environments by increasing the number of hazards from 10 to 15. In both environments, the aim of robots is to reach the goal position and have as few collisions with hazards as possible. We compare our approach (MBPPO-Lagrangian) with Unconstrained PPO [Schulman et al., 2017], model-free Safe RL approaches including Constrained Policy Optimization [Achiam et al., 2017], PPO-Lagrangian [Ray et al., 2019] and the model-based approach – safe-LOOP from Sikchi et al. [2021a].

The code for our approach is available here[2]. We run each algorithm with 8 random seeds for 450K environment interactions. The hyper-parameter settings and other experimental details are given in Appendix A. The performance of our approach and baseline algorithms is shown in Figure 2 for PointGoal environment and Figure 3 for CarGoal environment.

---

[1] https://github.com/openai/safety-gym
[2] https://github.com/akjayant/mbppol

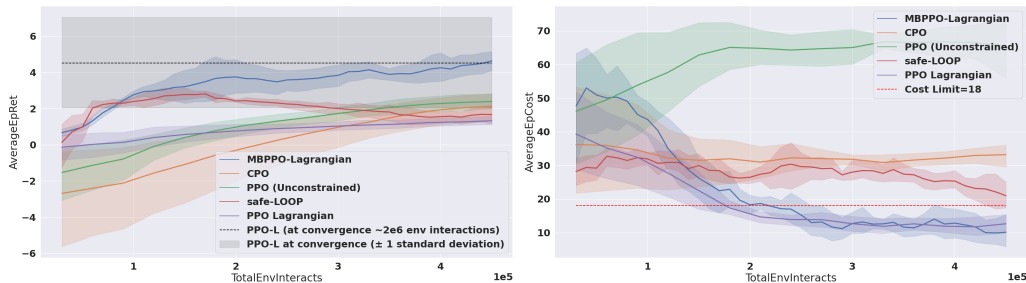

Figure 2: Reward Performance (Left) and Cost Performance (Right) in PointGoal Environment, where y-axis denotes Average Episode Reward Returns (left) / Cost Returns (right) and x-axis denotes total environment interacts

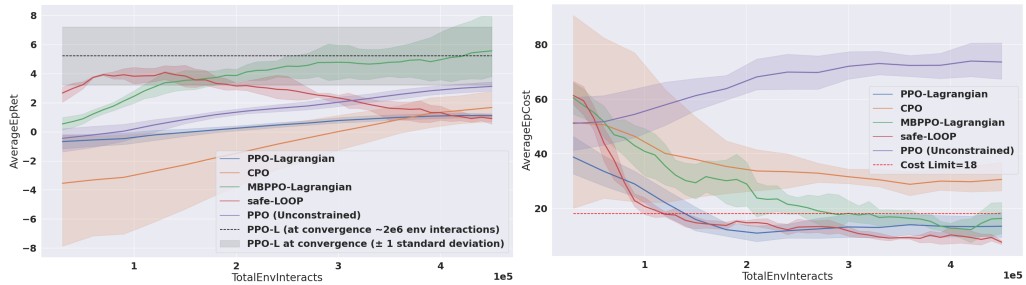

Figure 3: Reward Performance (Left) and Cost Performance (Right) in CarGoal Environment, where y-axis denotes Average Episode Reward Returns (left) / Cost Returns (right) and x-axis denotes total environment interacts

The left part of the plots represent Average Episode Return (average of the episode rewards) on y-axis and total environment interactions on the x-axis. The dashed lines represent performance of the PPO-Lagrangian (in black) at convergence which is around 2 million environment interactions. The right part of the plots show Average Episode Cost (average of the episode costs) on y-axis and total environment interactions on the x-axis. The red dashed line in this plot represents cost limit of 18. From the plots we can observe that our approach (MBPPO-Lagrangian) converges to the same level of reward performance as PPO - Lagrangian in just 450K environment interactions and outperforms the model-based safe-LOOP algorithm [Sikchi et al., 2021a]. Also our approach gives constraint adhering policies in both the tasks. In addition to above, we run our algorithm for various values of $\beta$ for both *CarGoal* and *RC-Car*[Ahn, 2019] in a similar manner as we did for *PointGoal* in Figure 1 and present it in Appendix D. We also compare our algorithm with safe-LOOP[Sikchi et al., 2021a] on *RC-Car* environment and present it in Appendix D as well. We found both approaches competitive in *RC-Car* environment, although safe-LOOP [Sikchi et al., 2021a] exhibits higher variance.

We also measure cumulative hazard violations that occur till convergence for MBPPO-L (ours), PPO-Lagrangian [Ray et al., 2019], safe-LOOP [Sikchi et al., 2021a] as follows -

$$Cumulative\ Violations = \sum_{Till\ convergence} \left[ \mathbb{1}(C(s_t, a_t) == 1) \right]. \tag{27}$$

We use *rliable* library (See Sec 4.3 in Agarwal et al. [2021] for more details) for plotting 95% confidence intervals for cumulative violations in the right part of Figure 4 (normalized by cumulative violations in unconstrained PPO in respective tasks) and final policy reward performance in the left part of Figure 4 (normalized by reward return of the final policy in unconstrained PPO in respective tasks) using mean, median and inter-quartile mean as aggregate estimates for our approach, PPO-Lagrangian and safe-LOOP because these approaches only give constraint satisfying policies at convergence. From figure 4 we can observe that our approach is competitive with regards the model-based safe-LOOP approach [Sikchi et al., 2021a] in terms of cumulative constraint violations but achieves better reward performance at convergence.

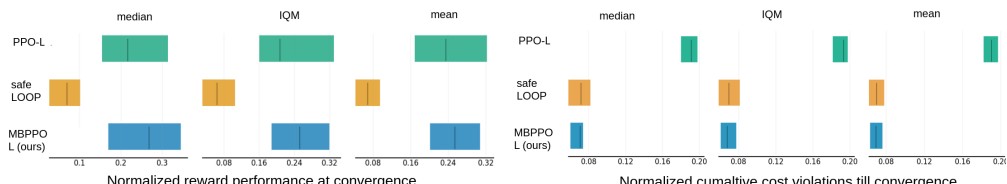

Figure 4: Normalized Reward Returns at Convergence (left) with median, inter-quartile mean (IQM), mean estimates and Normalized Cumulative Violations (right) with median, inter-quartile mean (IQM), mean estimates. Top rows (in green) represent PPO-Lagrangian, middle rows (in orange) represent safe-LOOP and bottom rows (in blue) represent our approach.

## 7  Conclusions

We presented a sample efficient approach for Safe Reinforcement Learning and compared our approach with relevant model-free and model-based baselines [Achiam et al., 2017, Sikchi et al., 2021a, Ray et al., 2019, Schulman et al., 2017]. Note that we chose the above baselines because they are specifically designed to solve a constrained optimization problem of the same structure as (1)-(3). In Sikchi et al. [2021a], approximation of the value function is used to provide long-term reasoning instead of using a truncated horizon. A limitation of this approach lies in the fact that approximations of reward and cost value functions are used. The most challenging part of model-based approaches is to learn the environment model. The first issue is of computational resources and the time overhead that is needed. We provide a comparison of the running time of algorithms and their hardware requirements (see Appendix B). Our algorithm does much better than safe-LOOP [Sikchi et al., 2021a] in terms of running time.

One should note that in safe exploration settings, agents do not explore as much as an unconstrained agent would do. This adds to the complexity of model learning in safe exploration problems because constrained exploration leads to limited representation of data points for model learning even with the same sample size. We found this more pronounced in high-dimensional environments like *DoggoGoal1*[Ray et al., 2019] (See Appendix C). Our algorithm depends on Lagrangian-based approach which provides a very simplistic way to construct a safe RL algorithm but suffers from the issue of low reward performance compared to unconstrained approaches (see Appendix A for a reward comparison). Note that we can have a similar model-based approach using TRPO-Lagrangian [Ray et al., 2019] but it involves an extra overhead of approximating the Fisher Information Matrix (FIM) and hence we chose a PPO-based approach. Also TRPO-Lagrangian and PPO-Lagrangian have similar performance on Safety Gym benchmark [Ray et al., 2019]. Increasing reward performance of Lagrangian-based approaches and devising better ways for model-learning in high-dimensional state representations in safe RL settings where exploration is limited, can be looked at in the future. Moreover, it would be interesting to adapt off-policy natural actor-critic algorithms such as in Bhatnagar et al. [2009], Diddigi et al. [2022] to the setting of constrained MDPs and study their performance both theoretically and empirically.

## Acknowledgments and Disclosure of Funding

Ashish K. Jayant was supported in his work through an MHRD scholarship from the Ministry of Education, Government of India; Indian Institute of Science, Bangalore; and Flipkart Internet Pvt. Ltd., Bangalore. S. Bhatnagar was supported in his work through the J.C.Bose National Fellowship, Science and Engineering Research Board, Government of India; a project from the Department of Science and Technology under the ICPS program; a project from DRDO under JATP-CoE; as well as the Robert Bosch Centre for Cyber Physical Systems, Indian Institute of Science, Bangalore.

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
