# A Hyper-parameters and finer experimental details

The hyper-parameters used for our algorithm are shown in Table 1. Each episode runs for 750 steps

Table 1: Hyper-parameters used for our algorithm MBPPO-Lagrangian

| Hyperparameter | Value/Description |
| --- | --- |
| No. of models in ensemble ($n$) | 8 |
| Hidden layers in single ensemble NN | 4 |
| (Hidden layers in single ensemble NN : no. of nodes) | (200:200:200:200) |
| Hidden layers in Actor NN | 2 |
| (Hidden layers in Actor NN : no. of nodes) | (64:64) |
| Hidden layers in Critic NN | 2 |
| (Hidden layers in Critic NN : no. of nodes) | (64:64) |
| Gradient Descent Algorithm | ADAM |
| Actor Learning Rate (for $\theta$ update) | 3e-4 |
| Critic Learning Rate (for $\psi_r, \psi_c$ update) | 1e-3 |
| Lagarange Multiplier Learning Rate (for $\lambda$ update) | 5e-2 |
| Initial Lagrange Multiplier Value ($\lambda_0$) | 1 |
| Cost limit ($d$) | 18 |
| Activation Function | $tanh$ |
| Discount Factor ($\gamma$) | 0.99 |
| GAE parameter ($\lambda$) | 0.95 |
| Horizon ($H$) in step 7 of Algorithm 1 | 80 |
| Validation Dataset/Train Dataset | 10%/90% |
| PR Threshold | 66% |
| Mix of real data and imaginary data for first pass | 5%95% |
| $\beta$ used in (29) | 0.02 |

as opposed to 1000 steps in the original version of Safety Gym, hence we chose cost threshold of 18 which is $\sim 75\%$ of 25 used in the official Safety Gym benchmark paper. In Goal-based Safety gym environments the aim is to reach the goal position (in green in Figure 1) with as few collisions as possible with hazards. If robot (in red in Figure 1) accesses 'hazard' positions (in blue in Figure 1), the agent incurs a cost = 1. The 'Point' robot has steering and throttle as action space while 'Car' robot has differential control. We use Performance Ratio (PR) threshold of 66%. The logic behind this number is that our agent should perform better in more than $50\%$ of the models in the ensemble. In our case we train an ensemble of 8 models out of which we use the best 6 models with minimum validation losses to calculate PR. We want our agent to perform better in at least 4 out of the 6 models that gives us the value as $66\%$. As mentioned in the text we increase the number of hazards in PointGoal1 and CarGoal1 environments from 10 to 15 so as to increase their difficulty level. We run on 8 random seeds which is greater than 4 used in benchmarking previous unconstrained model-based approaches [1]. We use the official implementation of baselines from safe-LOOP[2], CPO[3] and PPO-Lagrangian [4], respectively. In all baselines we use the same size of the neural networks as mentioned in Table 1 and the same number of models in ensemble for model learning for safe-LOOP. In safe-LOOP, we do not change the originally used planning horizon of $H = 8$ and cost threshold of 0 in the model optimization part. During first pass of Algorithm 1 (Line 5-17), we use $5\%$ of real environment interaction data and $95\%$ of imaginary rollout data, this is done to make use of real environment data that we collected to train model dynamics. After first pass, policy gets updated and after that we use purely imaginary data since this is an on-policy algorithm until PR goes below $66\%$.

The values used for normalization in plot for Figure 4 are the following: Reward performance of PPO-unconstrained = 20 and Cumulative constraint violations of PPO-unconstrained = 2e5.

---

[1] https://github.com/WilsonWangTHU/mbbl
[2] https://github.com/hari-sikchi/LOOP
[3] https://github.com/jachiam/cpo
[4] https://github.com/openai/safety-starter-agents

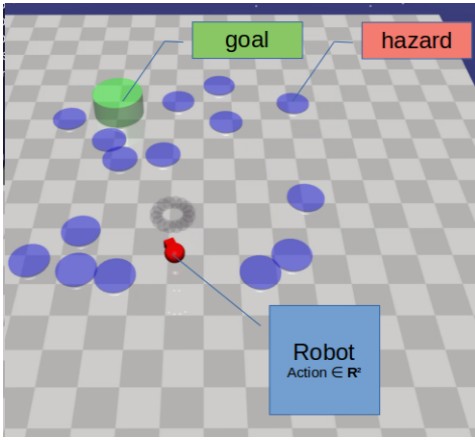

Figure 1: Safety Gym with labeled elements

## B Hardware requirements and Running time

Minimum 4 GB GPU space is required for running both the model based approaches. We did not

Table 2: Running time for 450k steps (in seconds)

| Algorithm | Running time (in s) |
| --- | --- |
| PPO-Lagrangian | $187.95 \pm 7.56$ |
| CPO | $266.25 \pm 6.46$ |
| MBPPO-Lagrangian | $21420.91 \pm 554.449$ |
| safe-LOOP | $183156.33 \pm 19083.43$ |

use multiprocessing in model-based approaches for data-collection from simulation. However, for model-free approaches we collected simulation data using 8 CPUs at a time. Running time for 450k steps for all baselines is given in Table 2.

Note that for convergence PPO-Lagrangian required $849 \pm 12.48$ s and CPO required $1162 \pm 8.87$ s. The running time of our model-based approach can be improved by using parallel workers for data collection but still major part of training is spent in model learning.

## C Comparison of model learning validation loss (Unconstrained vs Safe RL) and performance in High-dimensional task

We compare how model learning validation loss varies in Safe RL setting as opposed to unconstrained RL one. We plot validation loss vs no. of datapoints collected as our model-based algorithm progress (See Figure 2. We observe that validation error converges quickly in unconstrained setting, which is reflective of better exploration in unconstrained setting. This plot is generated using 8 different initialization of neural network used for model learning. We do this for our modified PointGoal1 environment.

*DoggoGoal1* is a challenging environment where a quadruped robot has to reach a goal position without colliding with obstacles. This task has higher dimensional action space (12-dimensional) than Point and Car robots which just needs steering and throttle in Safety Gym. On this task, even model-free Lagrangian baselines suffer and get quite low reward performance. Further, both safe-LOOP and MBPPO-Lagrangian fails to match model-free baseline in terms of reward performance in DoggoGoal environment. It signifies the additional challenge in model learning in case of high-dimensional environments.

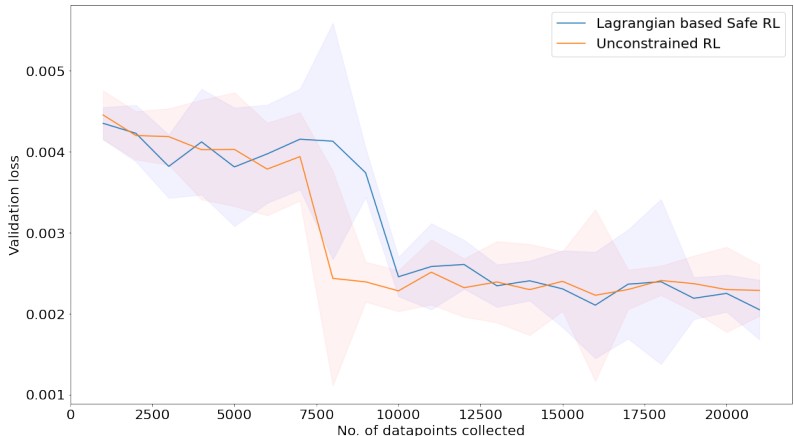

Figure 2: Quicker convergence of Validation loss in unconstrained setting as compared to constrained setting

Table 3: Reward Performance at convergence in DoggoGoal

| Algorithm | Episodic reward performance of final policy |
|---|---|
| PPO | $21.3 \pm 1.23$ |
| PPO-Lagrangian | $1.6275 \pm 0.46$ |
| safe-LOOP | $-0.14 \pm 0.05$ |
| MBPPO-Lagrangian | $-0.69 \pm 0.06$ |

# D    Effect of $\beta$ in CarGoal and RC-Car environment and Model-based baselines results on RC-Car

In RC Car[5] environment, a car has to rotate within a circle with target velocity for earning rewards. If car goes out of circle, it incurs cost.

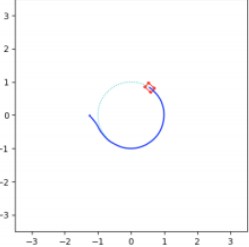

Figure 3: Car in (red) has to drive along the circle of fixed radius with some target velocity

---

[5]https://github.com/r-pad/aa_simulation

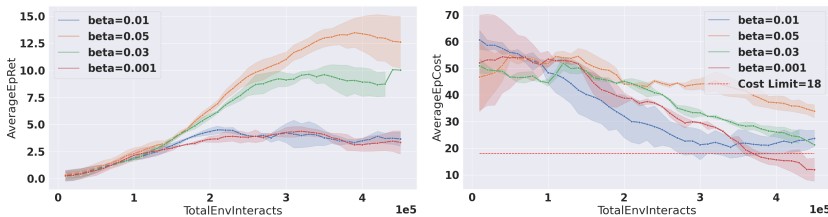

Figure 4: Effect of $\beta$ in CarGoal Environment on Reward returns (Left) and Cost returns (Right)

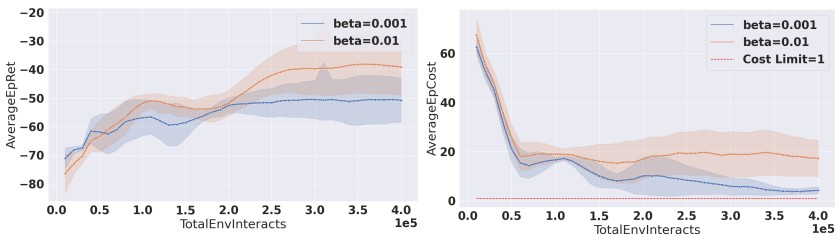

Figure 5: Effect of $\beta$ in RC-Car Environment on Reward returns (Left) and Cost returns (Right)

We observe a similar trend as observed in Figure 1 of the paper, i.e., lower the value of $\beta$, lower are the reward returns as the agent explores pessimistically. We also ran MBPPO-L(our algorithm) and safe-LOOP on the *RC-Car* environment for 150k steps (with 5 random seeds) only as safe-LOOP has a very high running time. We choose cost limit = 5. Results are presented in Figure 6. We can observe that both approaches have similar performance but safe-LOOP exhibits higher variance than our approach.

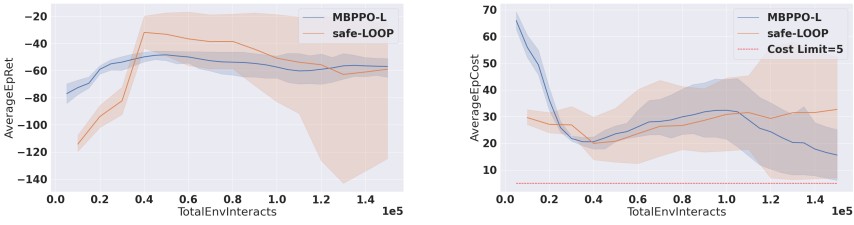

Figure 6: Reward returns (Left) and Cost returns (Right) in RC-Car environment