# OpenReview forum: "Model-based Safe Deep Reinforcement Learning via a Constrained Proximal Policy Optimization Algorithm"
_NeurIPS.cc/2022/Conference — NeurIPS 2022 Accept_

### Official Review · Reviewer_hwMj · 2022-07-10

**Rating:** 6
**Confidence:** 4
**Soundness:** 3 good
**Presentation:** 3 good
**Contribution:** 2 fair

**Summary:**

This paper proposes a safe reinforcement learning method in a model-based manner. The method uses the Lagrangian relaxation of the original constrained optimization problem and then uses dual gradient descent to find the saddle point. The method also improves the sample efficiency by learning an ensemble of dynamics models. Experiment results on Safety Gym environments demonstrate its effectiveness over model-free and model-based safe RL baselines.

**Questions:**

- About the experiment part, why does PPO-Lagrangian perform worse than MBPPO-Lagrangian asymptomatically? Since PPO-Lagrangian uses rollouts from the real environment while MBPPO-Lagrangian learns from model rollouts, the latter can have better sample efficiency but suffer from biases. So I was wondering why MBPPO-Lagrangian can outperform PPO-Lagrangian in terms of asymptomatic performances.

**Strengths And Weaknesses:**

### Strength
- Proposed method uses an ensemble of dynamics models to improve sample efficiency and address the problem of epistemic uncertainties.
- Experiment results show that MBPPO-Lagrangian significantly outperformstones in Safety Gym environments.
- It found that the safety criteria should be tighter in order to achieve better performance when using truncated model rollouts.

### Weakness
- It might be better if the evaluation part could be more comprehensive since only the results of two tasks are reported.
- The authors use dual gradient descent to find the local saddle point of the Lagrangian relaxation of the original constrained problem. It would be better if they can provide some mathematical analysis of the effectiveness of this approach.

---

> ### Author Response · Authors · 2022-08-02
> **Response to the reviewer**
>
> We thank the reviewer for taking the time to review our work! We address the concerns raised by the reviewer as follows:
>
>
> **"About the experiment part, why does PPO-Lagrangian perform worse than MBPPO-Lagrangian asymptomatically?"**
>
> Thanks for pointing this out, we have now updated the plots with variance of asymptotic performance of the PPO-Lagrangian (shaded grey area in both plots in Figure 2) to represent the $\pm 1$ standard deviation of the final policy performance of PPO-Lagrangian. We hadn't shown the standard deviation in the previous plots for the case of the PPO-Lagrangian. Also, in figure 4, we can observe that the 95% confidence bands of the reward performance of PPO-Lagrangian and MBPPO-Lagrangian are highly overlapping.

---

> > ### Comment · Reviewer_hwMj · 2022-08-07
> > **Thank you for the feedback.**
> >
> > Thanks to the authors for their feedback. Combined with other reviewers' comments, I think the current experiment evaluation is still limited, thus I will keep the same score.

---

### Official Review · Reviewer_Wj7m · 2022-07-11

**Rating:** 7
**Confidence:** 4
**Soundness:** 3 good
**Presentation:** 3 good
**Contribution:** 3 good

**Summary:**

The authors propose a new algorithm for a model-based safe RL (under the framework of CMDP). The proposed variant is modification of the PPO algorithm, namely by learning a model of the environment and introducing lagrangian relaxation to make sure the policy satisfies the safety constraints.


**Questions:**

How was the PR > 70% choice made? Did you run ablation analysis on this parameter?

Can you please run Figure 1 for more environments?

**Limitations:**

Adressed.

**Strengths And Weaknesses:**

The paper is well written and easy to understand. The main algorithm - Algorithm 1 - is also very helpful.

The motivation is sound, the results are interesting, and the algorithm itself seems easy to implement.

What I find as a satisfying sanity check is the fact that unconstrained PPO does reach the same or better performance as MBPPO-Lagrangian. I understand that as the graphs (Figure 2 and 3) use as x-axis the number of environment interactions, the model-based approach is likely to do better initially. But as we keep going, PPO really needs to get there, especially as it is unconstrained and it clearly heavily violates the constraints.


Weaknesses

> [266] CarGoal1 by increasing the number of hazards from 10 to 15.

  I am not a fan of this.. It’s fine if that’s another environment in the experiments, but the original unmodified environment should also be included.

PR > 70% is a bit of an arbitrary choice, and I don’t see this being discussed anywhere.

Due to the truncated horizon, the introduced \beta parameter and associated Equation (26) is something that is likely to be domain-specific. I would appreciate it if the authors could run Figure1 for multiple environments to see how the effect of beta varies across domains.

All the Figures, especially Figure 1 and Figure 4  are very hard to read, please increase the font size.

---

> ### Author Response · Authors · 2022-08-02
> **Response to the reviewer**
>
> We thank the reviewer for taking the time to review our work! We address the concerns raised by the reviewer as follows:
>
> **"How was the $PR > 70$ % choice made? Did you run ablation analysis on this parameter?"**
> The concept of Performance ratio was introduced in Kurutach et al.[1]  where they used 70% as the threshold. In our experiments, we use PR of 66% as mentioned in our supplementary material. We have updated line 5 in Algorithm 1 as "Performance ratio > PR Threshold" for the general use case. Thanks for pointing this out! The logic behind this number is that our agent should perform better in more than 50% of the models in the ensemble. In our case we train an ensemble of 8 models out of which we use the best 6 models with minimum validation losses to calculate PR. We want our agent to perform better in at least 4 out of the 6 models that gives us the value as $\sim66 $%.
>
>
>
> **"Can you please run Figure 1 for more environments?"**
>
> In the case of Safety Gym, even in model-free baselines such as PPO-Lagrangian, CPO, we observe that lower cost violations lead to lower reward performance since the agent explores pessimistically. With a lower value of beta (stricter cost threshold), we observe a lower reward performance in the case of the CarGoal environment as well. We have added a similar plot for the CarGoal environment in Appendix D of supplementary material. Other than safety gym, we have included a similar plot for the RC-Car [2] environment in Appendix D, where a car has to rotate within a circle with a certain target velocity for earning rewards. If the car goes out of the circle, it however incurs a cost. We observe a similar trend here, where lower beta, i.e., stricter threshold, leads to lower reward returns. The difference in the cost returns is also visible.
>
> **Presentation** : We have increased the font size of the text used in the figures.
>
> [1] Kurtland Chua, Roberto Calandra, Rowan McAllister, and Sergey Levine. Deep reinforcement
> learning in a handful of trials using probabilistic dynamics models. Advances in neural information processing systems, 31, 2018.
> [2] E. Ahn. Towards safe reinforcement learning in the real world. PhD thesis, 2019.

---

### Official Review · Reviewer_Y6ox · 2022-07-12

**Rating:** 5
**Confidence:** 4
**Soundness:** 3 good
**Presentation:** 3 good
**Contribution:** 2 fair

**Summary:**

The paper presents a model-based method for safe RL that learns; a dynamics model of the environment, a reward value function, a cost value function, and a policy. The method leverages the learned dynamic models to generate imaginary rollouts to learn the reward and cost value functions, which are then used by a lagrangian relaxation of PPO in the setting of constrained Markov decision processes.

The proposed method is evaluated in the Open AI safety Gym, where it achieved better rewards than constrained model-based baselines and it also obtained better sample efficiency with lower constraints violation than other constrained model-free approaches.

**Questions:**

- A static choice of the safety threshold seems very conservative. As training progresses I could imagine that the estimation of cost returns becomes less uncertain and thus relaxing the safety thresholds could allow for better performance in terms of rewards. How would you incorporate model uncertainty to set the safety threshold dynamically?

- Is there any particular reason why the method trains the value and cost function using purely imaginary rollouts? When new data is collected to learn new dynamics models, would there be any benefits of using such data to also improve the learned value and cost functions?

**Limitations:**

Yes. Limitations of the method are mentioned in the conclusions. The potential societal impact was qualified as Not Applicable by the authors.


**Strengths And Weaknesses:**

Strengths:
- The presented method is simple and builds on well-established methods. The results obtained a balance of the benefits of constrained model-based and model-free baselines,  showcasing better sample efficiency with low constraint violation wrt to constrained model-free approaches and better rewards wrt model-based methods.

- The paper is clear and well written. It presents a comprehensive related work and introduces a detailed background.

Weaknesses:
- The approach is evaluated in only two environments (PointGoal, CarGoal). Although the environments were modified to make them more challenging, the work would benefit from evaluating the method in more setups like the Doggo environment (which is partially reported in the appendix), or other variations of the Point and Car setting like the Button or Push setups from the Open AI Safety Gym.

- The method seems to struggle with longer time horizon tasks, due to the compounding errors of the learned dynamics, reward value, and cost functions. A conservative safety margin was introduced to deal especially with the underestimation of cost returns. However, such a safety margin results in a strong tradeoff wrt the reward performance.

---

> ### Author Response · Authors · 2022-08-02
> **Response to the reviewer**
>
> We thank the reviewer for taking the time to review our work! We address the concerns raised by the reviewer as follows:
>
> **"A static choice of the safety threshold seems very conservative. As training progresses I could imagine that the estimation of cost returns becomes less uncertain and thus relaxing the safety thresholds could allow for better performance in terms of rewards. How would you incorporate model uncertainty to set the safety threshold dynamically?"**
>
> We assume that the safety threshold is known and available (Refer L-145 of main paper) and hence we keep it static. For instance, think of a self-driving vehicle navigating in the face of uncertain traffic conditions. A safety threshold in this case could amount to having a safety bubble of say one metre around the car and if any vehicle comes within that distance, the car would then need to perform an action to save itself from a possible collision. Certainly a dynamically changing threshold would not make much sense in this case.
> Further, in our opinion, since agent explores pessimistically it is not guaranteed that as training progress, estimation of cost returns becomes less uncertain because due to its limited exploration agent might encounter states that it hasn't seen before. This phenomena should be more pronounced in an agent which explores pessimistically as opposed to an unconstrained agent.
>
>
> **"Is there any particular reason why the method trains the value and cost function using purely imaginary rollouts? When new data is collected to learn new dynamics models, would there be any benefits of using such data to also improve the learned value and cost functions?"**
> Thanks for pointing this out! For the first pass through the while loop (Algorithm 1, Line 5), we do mix real environment interactions with imaginary rollouts after which the policy gets updated.  Then for further passes until the PR doesn't degrade we use imaginary rollouts to update policy because real environment rollouts were collected from a different policy. We have now included a comment on this after line 6 of Algorithm 1 in the paper.
>
> **Experiments** : Based on the reviewer's comments, we have also run additional experiments in the limited time we had on the RC-Car [2] environment for different values of $\beta$.
> In this task, a car has to rotate within a circle with a certain target velocity for earning rewards. If the car goes out of the circle, it however incurs a cost.
> We have obtained initial results and have included these for now in Appendix D so that the reviewers can have a look at these additional results.
> The model-based baseline safe-LOOP [3] however has a high running time as mentioned in Appendix C and also required code modification before we could implement the same.
>
> [1] Harshit Sikchi, Wenxuan Zhou, and David Held. Learning off-policy with online planning. CoRL, 2021.
>
> [2] E. Ahn. Towards safe reinforcement learning in the real world. PhD thesis, 2019.

---

> > ### Author Response · Authors · 2022-08-04
> > **Typo in above response**
> >
> > Wrong reference number, safe-LOOP [3]  is  actually safe-LOOP[1].

---

### Official Review · Reviewer_CLz8 · 2022-07-25

**Rating:** 5
**Confidence:** 3
**Soundness:** 3 good
**Presentation:** 2 fair
**Contribution:** 3 good

**Summary:**

This paper presents Model-based PPO-Lagrangian (MBPPO-Lagrangian) algorithm for safe RL, which reduces epistemic and aleatoric uncertainty with an ensemble of neural networks and solves the underestimation problem of cost returns in a truncated horizon with the stricter threshold using a hyperparameter. The authors compared the algorithm in Safety Gym: PointGoal1 and CarGoal1. MBPPO-Lagrangian showed higher cumulative reward and lower cumulative cost performance than baseline methods.

**Questions:**

<Explanation of experiments>

- Please explain the environments you used. Also, please explain why you used PointGoal1 and CarGoal1 with more hazards, not PointGoal2 and CarGoal2.
- The names ``PointGoal`` and ``PointGoal1`` both seem to be used to name the same environment (e.g. Figure 2), so please unify the name. The same goes for the CarGoal1 environment too.
- Figure 4 is hard to interpret. I cannot understand what you want to say from Figure 4. Is there any reference that made you decide to show the data in this format?

<More baselines>

I think the comparison between yours and Liu et al.[1] should be done since you mentioned the common points in L264. Also, Liu’s paper dealt with the same environment as yours and they are also model-based.

<Details>

- L163: is $\rightarrow$ in
- I think $t+1$ should be $t+l$ in Eq. 17 and Eq. 18.

[1] Zuxin Liu, Hongyi Zhou, Baiming Chen, Sicheng Zhong, Martial Hebert, and Ding Zhao. Safe model-based reinforcement learning with robust cross-entropy method, 2020.


**Limitations:**

Yes, the authors mentioned the limitation at the end of the conclusion.

**Strengths And Weaknesses:**

The authors successfully applied a model-based approach to the PPO-Lagrangian method by reducing the uncertainty of state transition using an ensemble of neural networks. Also, they suggested a simple solution that uses a hyperparameter to fix an innate underestimation-of-cost problem in the model-based approach that assumes a truncated horizon.

However, I suggest the authors strengthen the Experiments part. The environments used in experiments are not sufficiently explained and I feel the paper needs more baseline experiments. The more detailed suggestions will be described in the next section.

---

> ### Author Response · Authors · 2022-08-02
> **Response to the reviewer**
>
> We thank the reviewer for taking the time to review our work! We address the concerns raised by the reviewer as follows:
>
> **"Please explain the environments you used. Also, please explain why you used PointGoal1 and CarGoal1 with more hazards, not PointGoal2 and CarGoal2"** :
>
> Safety Gym consists of several environments with choice of robot and difficulty of tasks. We test all baseline algorithms and
> our work on our modified version of PointGoal and CarGoal environments where robots are 'Point' and 'Car' and task correspond to 'Goal'. Please refer to the screenshot of PointGoal environment with labeled robot, hazard and goal as shown in Figure 1 of the supplementary material (SM). In Goal-based environments the aim is to reach the goal position (shown in green in Figure 1 of SM) with as few collisions as possible with hazards. If robot (shown in red in Figure 1 of SM) accesses 'hazard' positions (in blue in Figure 1 of SM), the agent incurs a cost = 1. We modify the original environment to remove vases (a fragile box-type obstacle) because cost of touching a vase is function of velocity of the vase after collision (Reference : https://github.com/openai/safety-gym/blob/master/safety_gym/envs/engine.py)
> which is not part of the robot's state vector. The difference between PointGoal1 and PointGoal2 is more number of vases, we compensate that with increasing number of hazards.
> We apologise for the confusion regarding names and will unify the names "PointGoal1" and "PointGoal" and similarly for CarGoal environments.
>
> **"Figure 4 is hard to interpret. I cannot understand what you want to say from Figure 4. Is there any reference that made you decide to show the data in this format?"** :
> We plot figure 4 using 'rliable' library by Agarwal et al. [2] which uses statistical techniques to provide a more robust way of comparing RL algorithms in order to deal with statistical uncertainty. We normalise scores in the respective tasks by dividing the reward performance of the final policy by the reward performance of the final policy of unconstrained PPO in that task so that the performance can be compared across tasks as well. Similarly we do this for cumulative cost violations as well. Then we plot 95% confidence intervals of mean, median, inter-quartile mean estimates computed across different runs (or seeds) and tasks. Please refer to Agarwal et al. [2] for more details of their approach. From Figure 4, we would like to convey that across all estimates (mean/median/IQM), our approach is better than other model based approaches by Sikchi et al. [3] in terms of reward performance (left) while it is competitive in terms of cumulative cost violations (right) since 95% confidence intervals are overlapping.
>
>
> **"I think the comparison between yours and Liu et al.[1] should be done ... model-based."**: We do not benchmark Liu et al. [1] for two main reasons -- first, their optimisation problem structure consists of single-stage cost constraint while ours is across a finite horizon (See eq 1 of Liu et al. [1]) and second, they initially train their environment model by collecting random episodes for 50,000 steps as can be seen from their code (Refer line 62,63 of https://github.com/liuzuxin/safe-mbrl/blob/master/run.py) and their implementation doesn't account for hazards violations caused in that phase. Also their implementation counts three cost violations as 1 which underestimates risk of their approach (See line 117-135 of https://github.com/liuzuxin/safe-mbrl/blob/master/utils/env_utils.py, see also line 16 of https://github.com/liuzuxin/safe-mbrl/blob/master/run.py).
> Thanks for pointing out typos in  L163 and Eq 17,18, we have corrected them in the paper.
>
> **Experiments** : Based on the reviewer's comments, we have also run additional experiments in the limited time we had on the RC-Car [2] environment for different values of $\beta$.
> In this task, a car has to rotate within a circle with a certain target velocity for earning rewards. If the car goes out of the circle, it however incurs a cost.
> We have obtained initial results and have included these for now in Appendix D so that the reviewers can have a look at these additional results.
> The model-based baseline safe-LOOP [3] however has a high running time as mentioned in Appendix C and also required code modification before we could implement the same.
>
> [1] Zuxin Liu, Hongyi Zhou, Baiming Chen, Sicheng Zhong, Martial Hebert, and Ding Zhao. Safe model-based reinforcement learning with robust cross-entropy method, 2020.
>
> [2] Rishabh Agarwal, Max Schwarzer, Pablo Samuel Castro, Aaron C Courville, and Marc Bellemare. Deep reinforcement learning at the edge of the statistical precipice. Advances in Neural Information 326 Processing Systems, 34, 2021.
>
> [3] Harshit Sikchi, Wenxuan Zhou, and David Held. Learning off-policy with online planning. CoRL, 2021.
>
> [4] E. Ahn. Towards safe reinforcement learning in the real world. PhD thesis, 2019.

---

> > ### Comment · Reviewer_CLz8 · 2022-08-10
> > **Thank you for the feedback.**
> >
> > Thanks for addressing all my concerns regarding the paper!
> >
> > I have two more recommendations to improve the paper. I wonder whether showing the standard deviations of graphs in Figure 1 and in Appendix D is possible. Also, it would be better to size up figures and the fonts in the figures. I saw the revised version, but the readability is still not improved.
> > To sum up, I will keep my score same, but I recommend adding more experiments as other reviewers also mentioned.

---

> > > ### Author Response · Authors · 2022-08-10
> > > **Thanks for the feedback**
> > >
> > > We thank you for your comments. Your suggested changes will be incorporated.

---

### Author Response · Authors · 2022-08-02
**Summary of changes to the paper**

We summarise below the broad changes made based on the reviewer comments.

1) We have included an explanation of the environments in appendix A.
2) We have included the variance of PPO-Lagrangian performance at convergence in plots of the main paper in Figures 2 and 3.
3) We have added an additional experiment which shows perfomance variation of the agent with respect to $/beta$ in Appendix D on the CarGoal and RC-Car environments.
4) As suggested by reviewers, we have unified the names PointGoal1 as PointGoal and CarGoal1 as CarGoal.
5) We have used a more general keyword "PR Threshold" instead of 70% in Algorithm 1 earlier and included comments on PR threshold in Appendix A.
6) We have included a comment on the usage of real and imaginary data in the Algorithm 1 (after L6) and explained the same in Appendix A as well.
7) We have corrected the typos in L163 and eq-17,18.

---

### Meta-Review · Area_Chair_rAcs · 2022-08-26

**Recommendation:** Accept
**Confidence:** Certain

**Metareview:**

This paper presents Model-based PPO-Lagrangian (MBPPO-Lagrangian) algorithm for safe RL, which reduces epistemic and aleatoric uncertainty with an ensemble of neural networks. The authors experimented the proposed algorithm in safety benchmarks such as  Safety Gym: PointGoal1 and CarGoal for which MBPPO-Lagrangian showed better performances and safety guarantees than other model-free/based safe RL baseline algorithms. This paper presented a model-based safe RL algorithm that has immense applications in RL for safety-critical problems. The paper is generally well-written and intuitive, with most concepts clearly explained. The safety results demonstrated in the experiments are convincing and the algorithms are easy enough to implement for most practical applications. Therefore, the review committee has a consensus of recommending acceptance for this work to NeurIPS 22.



**Award:**

No

---

### Decision · Program_Chairs · 2022-09-14

Accept